# Sunlight-Mediated Green Synthesis of Silver Nanoparticles Using the Berries of *Ribes rubrum* (Red Currants): Characterisation and Evaluation of Their Antifungal and Antibacterial Activities

**DOI:** 10.3390/molecules27072186

**Published:** 2022-03-28

**Authors:** Humaira Rizwana, Mona S. Alwhibi, Rawan A. Al-Judaie, Horiah A. Aldehaish, Noura S. Alsaggabi

**Affiliations:** Department of Botany and Microbiology, College of Science, King Saud University, P.O. Box 22452, Riyadh 11495, Saudi Arabia; malwhibi@ksu.edu.sa (M.S.A.); 438202769@student.ksu.edu.sa (R.A.A.-J.); haldehish@ksu.edu.sa (H.A.A.); nalsaggabi@ksu.edu.sa (N.S.A.)

**Keywords:** sunlight, silver nanoparticles, *Ribes rubrum* (red currant), antifungal activity, antibacterial activity

## Abstract

Plants are a treasure trove of several important phytochemicals that are endowed with therapeutic and medicinal properties. *Ribes rubrum* L. (red currants) are seasonal berries that are widely consumed for their nutritional value and are known for their health benefits. Red currants are a rich source of secondary metabolites such as polyphenols, tocopherols, phenolic acids, ascorbic acid, and flavonoids. In this study, sunlight-mediated synthesis of silver nanoparticles (AgNPs) was successfully accomplished within 9 min after adding the silver nitrate solution to the aqueous extract of red currant. The synthesised AgNPs were characterised with UV–Vis, transmission electron microscopy (TEM), dynamic light scattering (DLS), Fourier transform infrared spectrum (FTIR), and energy-dispersive X-ray spectrum (EDX). The efficacy of aqueous extracts of red currants and AgNPs in controlling the growth of some pathogenic fungi and bacteria was also investigated. The UV–visible (UV–Vis) spectrum displayed an absorption peak at 435 nm, which corresponded to the surface plasmon band. The strong silver signal on the EDX spectrum at 3 keV, authenticated the formation of AgNPs. The several peaks on the FTIR spectrum of the aqueous extract of red currant and the nanoparticles indicated the presence of some important functional groups such as amines, carbonyl compounds, and phenols that are vital in facilitating the process of capping and bioreduction, besides conferring stability to nanoparticles. The TEM microphotographs showed that the nanoparticles were well dispersed, roughly spherical, and the size of the nanoparticles ranged from 8 to 59 nm. The red currant silver nanoparticles were highly potent in inhibiting the growth and proliferation of some fungal and bacterial test isolates, especially *Alternaria alternata*, *Colletotrichum musae*, and *Trichoderma harzianum*. Based on the robust antifungal and antibacterial activity demonstrated in this study, red currant nanoparticles can be investigated as potential replacements for synthetic fungicides and antibiotics.

## 1. Introduction

Nanotechnology is a remarkable and evolving technological innovation that has fascinated many scientists from multidisciplinary fields. The distinctive properties of nanoparticles (NPs) have facilitated their fabrication to nanoscale levels that are applicable to innumerable applications in biomedicine, pharmaceuticals, chemical engineering, and agriculture [1,2,3]. Recently, nanoparticles have been referred to as “revolutionising materials”, as they are endowed with unique physicochemical properties such as nanosize, surface charge, surface volume ratio, morphology, and stability, besides possessing the ability to modify mechanical, catalytic, and thermal properties [4,5,6,7]. Amongst the many tasks executed by the NPs, DNA probing, gene delivery, drug delivery, protein detection, and biological labelling are the most important [2,3,4,8]. Similarly, smart outcomes have also been achieved in the agricultural sector.

Several metals and their oxides are used to create nanoparticles. Nanoparticles are widely used in medical, pharmaceutical, and agricultural fields due to their ability to deliver drugs rapidly in a controlled manner, as well as their good dissolution, absorption, bioavailability, and diagnosis [9,10,11,12]. Among gold, silver, iron, zinc, and copper, silver is the most desirable metal used in the creation of nanoparticles, mainly due to its adaptability, conductivity, versatility, and optical and electrical properties [7,13]. Additionally, silver has been known for its antimicrobial properties for ages. Hence, it has been explored for its therapeutic value both as silver-based formulations and as silver ions [14,15]. Silver is also associated with antibacterial, antifungal, and anti-inflammatory properties [13,16]. Chemical, physical, and biological methods are often employed in the synthesis of NPs, both on a large and small scale [17]. Although chemical and physical methods of nanoparticle synthesis are still widely used, they have some drawbacks, such as the use of toxic chemicals and solvents, the formation of hazardous by-products, and low yield and high cost [17,18]. In contrast, biological synthesis is environmentally friendly, sustainable, and non-hazardous. Biological synthesis of NPs is carried out with algae [19], fungi [20], bacteria [21], and plants [22]. Nevertheless, again, among all of the biological methods, plant-mediated silver nanoparticle synthesis is the most preferred.

Plants are a treasure trove of several important phytochemicals that are bestowed with therapeutic and medicinal properties. Additionally, plants are abundant in nature, and several of them have not been explored for their immense beneficial effects, especially as nanoformulations in the agricultural, medicinal, and pharmaceutical industries. Hence, plant-mediated silver nanoparticle synthesis is a superior alternative solution to the other biological methods of synthesis, as it does not require the laborious preparation of culture media and also saves the effort, time, and cost spent on their maintenance [23]. Most importantly, plant-based nanoparticle synthesis is quick, safe, economical, non-hazardous, sustainable, biocompatible, and reliable [2,8].

Green synthesis of NPs uses plant material and is also referred to as phyto-fabrication. Synthesis of silver nanoparticles (AgNPs) has been reported from almost all parts of plants, including seeds and seed waste [24,25], arils [26], fruit and fruit waste [27,28], flowers [29], stems [30], roots [31] and leaves [32]. The green synthesised nanoparticles have shown potent cytotoxic, antifungal, and antibacterial activity against different cancer cell lines and several pathogenic isolates of bacteria and fungi [21,24,26]. Recently, AgNPs derived from aqueous extracts of mace arils showed strong inhibition of *Fusarium oxysporum*, *Alternaria alternata*, and *Trichoderma harzianum*, along with a few bacterial species [26]. AgNPs synthesised from seed extracts of *Alpinia katsumadai* and leaf extracts of *Trigonella foenum*-*graecum* L. and *Thymbra spicata* L. var. spicata showed excellent cytotoxic and antibacterial activity against some bacterial isolates [24,32,33]. Similarly, *Abelmoschus esculentus* flower extract demonstrated potent antiproliferative and cytotoxic activity; several Gram-positive and Gram-negative bacteria were also significantly inhibited [29]. The robust activity of green synthesised AgNPs against microorganisms has been attributed to the innumerable bioactive compounds present in plant extracts that contribute towards the capping, reduction, and stability of NPs during the synthesis [34,35].

With the upsurge in resistant bacterial and fungal pathogens in the last decade, there is an increasing need for antimicrobial compounds that can combat resistant strains [36,37,38]. Many plant pathogenic fungi cause massive post-harvest losses to crops, especially during transport, storage, and marketing. In order to reduce these losses, indiscriminate use of synthetic fungicides has led to resistant strains of several fungal species. Similarly, bacterial resistance has also been exacerbated due to a lack of alternative antibiotics against the common resistant strains. Hence, in the present scenario, plant extract-mediated NPs would be superior alternatives or replacements to tackle the ever-growing concerns about antimicrobial resistance. With this notion in mind, we aimed to synthesise AgNPs from the berries of *Ribes rubrum* and investigate their antimicrobial properties against a panel of microorganisms.

*Ribes rubrum* (Rb) is a shrub that belongs to the Grossulariaceae family [39]. The sour-tasting, translucent, intense red, small berries of Rb are known for their immense health benefits and are commonly called red currants (Rc). They are consumed either fresh or as juices, jams, or processed food supplements [40]. Red currants possess a wide variety of phytocompounds and are rich in nutritional composition, including provitamins, vitamins (C, E, and A), organic acids, folic acid, minerals, phytosterols, and carbohydrates [41]. Secondary metabolites found in the berries include polyphenols, ascorbic acid, phenolic acids, and flavonoids [42,43]. Their consumption has been linked to improved dietary management of diseases such as cancer, hypertension, osteoporosis, and cardiovascular disease [44,45]. In addition, a few studies have also reported biological activities of Ribes species, such as anti-inflammatory [46], antioxidant [47], antitumor [48], and antimicrobial [41,49,50]. Based on the phytochemical composition and the wide range of biological activities shown in earlier studies, we aimed to synthesise AgNPs from red currants (Rc) and evaluate their efficacy against a panel of fungi and bacteria. Furthermore, to the best of our knowledge, red-currant-mediated AgNP synthesis and evaluation against fungal cells have not been reported previously.

## 2. Results and Discussion

### 2.1. Visual Examination of Nanoparticle Formation and UV–Vis Spectroscopy

The synthesis of NPs was carried out with red currants (berries) aqueous extract and silver nitrate in the presence of sunlight. The sunlight-driven NP synthesis was conducted following the method of Kumar et al. [51]. For the synthesis of NPs, a light pink aqueous extract of red currants was added to a fixed volume of a colourless silver nitrate (AgNO_3_) solution in a glass flask. The addition of extract to the AgNO_3_ solution led to a pale pink reaction mixture, which was immediately kept in direct sunlight, shaken occasionally, and the time taken to change the colour of the reaction mixture was noted. The colour of the reaction mixture started to change immediately after 1 min. A rapid transformation to a dark brown colour was observed at 9 min of sunlight exposure. The colour change ceased after 9 min, marking the end of the nucleation process or the complete reduction of silver ions to silver NPs (Figure 1). The time taken for the entire process of NP synthesis indicates that sunlight-assisted NP synthesis is a quick, cost-effective, and productive method of synthesis.

The visual colour transition (pink to brown), witnessed during the synthesis process in this study, strongly inferred the formation of Rc-AgPNs. However, the formation of NPs was further checked by analysing the synthesised NPs with a UV–Vis spectrophotometer. The absorption spectrum of Rc-AgNPs displayed a peak at 435 nm in the UV spectrum, which corresponded to the surface plasmon band (SPR) (Figure 2). It is a well-recognised fact that the excitation of SPR in metal NPs causes the reaction mixture to change its colour to brown [52]. The collective oscillations of free electrons present in NPs, along with the light waves, give rise to a characteristic peak between 410 and 500 nm [53,54]. Very close to the findings of this study, an LSPR peak at 430 nm was displayed by the AgNPs synthesised from the pomace of black currant [28].

The effect of electromagnetic radiation from sunlight in facilitating the formation of green NPs has been highlighted in previous studies [26,54]. Under direct sunlight, the reaction mixture of pomelo peel extracts and AgNO_3_ changed its colour immediately within a minute or less, which later turned dark brown, indicating the formation of AgNPs [55]. This is in agreement with the findings of the present study. Hence, photo-irradiated synthesis is a quick and easy method of synthesis. A recent study reported the rapid biosynthesis of AgNPs from mace aqueous extract in the presence of sunlight [26]. Some previous studies have put forth their views on the possible mechanisms of sunlight, sunlight intensities, and ultraviolet (UV) light-driven NP synthesis [56,57]. A previous study showed the sunlight-induced synthesis of AgNPs with citrus lemon extracts and suggested that the UV light in sunlight could be a possible source of light irradiation [58]. Nevertheless, another study showed the effect of different light filters on NP synthesis. Amongst the red, green, blue, and yellow lights, blue light was the most effective, suggesting that blue light plays a vital role in the reduction of Ag^+^ ions [59]. However, in the present study, the role of UV light from sunlight can be ruled out because the nanoparticle synthesis was carried out in a glass container and the transmission of UV light is totally diminished, as UV light cannot pass through the glass containers [60]. Hence, in the present study, the blue light of the visible spectrum of sunlight may have played a role in the reduction of Ag^+^ ions to metallic Ag. It has been suggested that the blue light from sunlight induces tautomerisation (enol to keto form) of biomolecules. The tautomerisation releases reactive hydrogen atoms, which could possibly cause the reduction of Ag^+^ ions [60].

### 2.2. FTIR Analysis (Rc Extracts and AgNPs)

Fourier transform infrared spectroscopy is an important tool that provides insight into the biomolecules present in a given sample. In this study, the Rc extract and synthesised Rc-AgNPs were subjected to FTIR analysis. The IR spectrum of Rc aqueous extract and Rc-AgNPs displayed peaks at different wavelengths that corresponded to biomolecules that aided in the capping and bioreduction during the synthesis of nanoparticles (Figure 3). The Rc-AgNPs and Rc extract displayed some peaks at similar positions on the spectrum, which denoted the following: strong and broad peaks related to OH stretches of alcohols and phenols (3404 cm^−1^, 3407 cm^−1^), medium peaks of C-H asymmetric stretching of alkanes (2931 cm^−^^1^, 2933 cm^−^^1^), sharp peaks related to the C=O stretching vibrations of aromatic compounds (1721 cm^−^^1^, 1724 cm^−^^1^), the weak peaks that denoted the presence of NH (amines) resulting from the stretching vibrations of the carbonyl group of proteins (1625 cm^−^^1^, 1626 cm^−^^1^); the medium sharp peaks of C-N (amine skeletal stretch) and C-O stretching vibrations of alcohols (1057 cm^−^^1^ and 1058 cm^−^^1^). Further, the IR spectrum of Rc-extract displayed two peaks (1406 cm^−^^1^ and 1232 cm^−^^1^), which corresponded to the stretching and bending vibrations of C-N (amines) and OH (alcohols). However, these peaks (1406 cm^−^^1^ and 1232 cm^−^^1^) did not appear on the IR spectrum of Rc-AgNPs. Instead, a new peak appeared at 1362 cm^−^^1^ on the IR spectrum of Rc-AgNPs. The disappearance of some peaks from the spectrum of Rc-AgNPs suggests the role of amines, carbonyl compounds, and phenols in capping and reduction (silver to AgNPs) during nanoparticle synthesis.

In accordance with the present findings, the disappearance of some bands in the IR spectrum of synthesised NPs from the pomace of black currant and apricot extracts was implied to be due to the capping action of carbonyl groups (amines) [61]. Biomolecules in plant extracts play vital roles in the capping and stabilisation of synthesised NPs. Phenols (OH group) and proteins (carbonyl group) have special affinities towards silver and other metals and form a layer that surrounds the NPs, resulting in capping [62,63]. Amines and alcohols also prevent agglomeration during the synthesis of NPs [64]. A recent study reported high amounts of phenolic acids in the pomace of red currant extracts [65]. In concurrence with many studies, hydroxyl and carboxylic acid are the two highly significant functional groups that are involved during the formation of NPs [66,67,68]. More specifically, during the formation of NPs, the carbonyl group (proteins) helps in stabilisation, while hydroxyl groups (alcohols/phenols) assist in the reduction process [65]. Similar to the findings of this study, the FTIR analysis of silver and selenium NPs synthesised from the pomace of black currant revealed the presence of several functional groups such as phenols, coumarins, carbonyl compounds, carboxylic acids, amines, and other important functional groups [28,61,69]. The phenols present in the extracts of black currant could have played a crucial role in the reduction, capping, and stabilisation of NPs during nanoparticle synthesis [61].

### 2.3. Energy-Dispersive X-ray Analysis (EDX)

Energy-dispersive X-ray analysis, coupled with field-emission scanning electron microscopy (SEM), revealed the elemental composition of Rc-AgNPs. The EDX spectrum displayed a strong peak at 3 keV, confirming the presence of elemental silver and its subsequent role in its reduction to AgNPs. Peaks between 2.7 and 3.5 keV are indicative of the formation of AgNPs, which arise from the SPR [62]. Several other elements that were relatively evident on the spectrum are as follows: Ca (0.2%), Mg (0.2%), Na (0.4%), Cl (1.7%), and K (2%). Carbon (52% and oxygen (23%) are also evident on the spectrum. In accordance with the findings of this study, the EDX spectrum of AgNPs synthesised from black currant pomace showed a typical optical absorption peak related to metallic silver at 3 keV [28,61]. In another study, the elemental composition of red currants showed different quantities of elements such as calcium, magnesium, manganese, zinc, phosphorus, iron, and potassium [70]. These findings are in agreement with the present study. However, another study showed the presence of carbon and oxygen on the EDX spectrum of green synthesised nanoparticles [71]. Hence, the different elements that are shown in the Rc-AgNP spectrum (Figure 4) could be the minerals present in the Rc extracts that surround the AgNPs during synthesis.

### 2.4. Size and Morphology Studies of Rc-AgNPs with TEM and DLS

The size of the NPs and their morphology are important parameters that govern the in vitro toxicity of NPs [72]. Hence, the determination of size and shape is vital in the characterisation of nanoparticles. Transmission electron microscopy and dynamic light scattering are the most widely used tools to determine morphological characteristics, shape, and size distribution. The Rc-AgNPs were screened with the help of a transmission electron microscope to understand the morphology and size of the NPs. The microphotographs thus obtained showed that the particles were roughly spherical, without any agglomeration. The size of the particles ranged between 4 nm and 59 nm (Figure 5). The synthesised Rc-AgNPs were further analysed with a zeta sizer through a process of dynamic light scattering (DLS). The DLS spectrum validates the size distribution and polydiversity index (PDI). Average size of 85 nm with a PDI of 0.17 was displayed on the spectrum (Figure 6). The slight disparity in the sizes shown in TEM and DLS could be due to the state in which they are measured. The DLS measurements are taken in a solvated or hydrated state, while the TEM operates in a dry state and are based on the number of electrons transmitted through the surface projected [73,74]. In a fluid phase, the NPs particles are in a constant state of movement due to Brownian motion [75]. In addition, the charge present on the NPs creates interaction between ions, molecules, and surfaces, which leads to the formation of adsorbed layers on the surface of the AgNPs [76,77]. Hence, the size of the NPs in DLS is not just the NP size but is inclusive of the hydro diameter of the biomolecules surrounding them, which are under the influence of Rayleigh scattering and Brownian motion [75,77,78]. Therefore, the measurements of TEM and DLS cannot be corroborated, and the measurements by DLS reported a generally larger size than those measured by TEM [74]. Similarly, DLS and TEM analysis of AgNPs synthesised from black currant pomace, revealed the spherical shape of NPs that measured about 40–60 nm [27,28,57,61]. Nevertheless, another study showed that the AgNPs synthesised from red currant waste were cubic, and their hydrodynamic diameter ranged between 25 nm and 65 nm [65].

### 2.5. Antifungal Activity of Red Currant Aqueous Extracts and Rc-AgNPs

Figure 7 depicts the antifungal activity of Rc-AgNPs against an array of plant pathogenic fungi. The synthesised Rc-AgNPs exhibited robust antifungal activity against all the test isolates except *P. mangiferae*. When treated with Rc-AgNPs, *Alternaria alternata*, *Colletotrichum*
*musae*, and *Trichoderma harzianum* showed the complete arrest of mycelial growth, indicating the significant impact on the proliferation of the mycelium (Figure 8 and Figure 9). As displayed in Figure 10, *Alternaria alternata*, *Colletotrichum musae*, and *Trichoderma harzianum* exhibited 100% mycelial growth inhibition, followed by *Fusarium oxysporum* (88%) and *Botrytis cinerea* (66%). However, poor inhibitory activity was witnessed with *Pestalotiopsis* *mangiferae* (41%). The altered colony morphology of *Botrytis cinerea* depicted in Figure 8 indicates poor sporulation. On the other hand, aqueous extracts of Rc showed weaker mycelial growth inhibition in comparison to Rc-AgNPs. Except for *Colletotrichum musae* (57%) and *Fusarium oxysporum* (46%), all the other fungal isolates showed poor inhibition when treated with Rc aqueous extracts (Figure 7, Figure 8 and Figure 9). The antifungal activity caused by AgNO_3_ on all the fungal isolates was rather negligible in comparison to Rc-AgNPs and Rc extract. The Rc-AgNPs inhibited almost all the test isolates, which was as significant as the inhibition caused by the fungicide. In a previous study, gold nanoparticles, synthesised using fruit extracts of *Ribes nigrum* (blackberry), were evaluated against many microorganisms, including *Aspergillus niger* and *Trichophyton rubrum*. Both the fungal isolates were inhibited, with a minimum inhibitory concentration (MIC) of 26 µg mL^−^^1^, while the minimum fungicidal concentration (MFC) values ranged between 26 to ≥52 µg mL^−^^1^ [79]. Another study reported significant growth inhibition of *Candida albicans* with extracts of four varieties of black currant (*Ribes nigrum*) at 0.4 to 250 mg mL^−^^1^ [41]. However, according to another report, leaves and branches of eight Ribes species, including *Ribes rubrum*, did not show any significant inhibitory effect on *Candida albicans* and *Candida parapsilosis* [80]. Berries and leaves of seven cultivars of blackberry were extracted with methanol and screened against several microorganisms, including *Aspergillus niger* and *Candida albicans*. Berries were more effective than leaves against both the above-mentioned fungi. The minimum inhibitory concentrations of berries and leaves against *Aspergillus niger* were 40–83 μg mL^−^^1^ and 94–157 μg mL^−^^1^, and were much lower than those for *Candida albicans* (56–87 μg mL^−^^1^ and 147–283 μg mL^−^^1^) [81].

The robust inhibitory activity of Rc-AgNPs shown in the present study could be due to their small size and their ability to permeate the cell and change its stability by disturbing the vital metabolic functions. This directly impacts the spore germination and proliferation of mycelium, leading to cell death. Similar to the present finding, a recent study suggested a possible mode of action of green synthesised nanoparticles against a plant pathogen, *Sclerotium rolfsii*. Based on their findings, the antifungal activity of green AgNPs could perhaps be due to the leakage of sugars and proteins from the outer membrane of the mycelium, resulting in disturbed permeability. The interaction of NPs with the fungal cell, especially after entering the inner membranes, causes inactivation of respiratory chain dehydrogenase and thereby limits respiration, which adversely affects mycelial growth. Additionally, the degeneration of the cellular membranes is also facilitated by phospholipids. Hence, the cumulative action of sugars, proteins, and phospholipids eventually causes the death of the fungal cell [82]. Various views have been presented to explain the possible antifungal mechanism of AgNPs. However, it is an accepted fact that NPs disturb cellular integrity. The entry of NPs into the fungal cell results in enhanced levels of ROS molecules (hydroxyl and peroxide radicals), which impairs RNA and DNA synthesis. Furthermore, the nanoparticle destroys the proteins and lipids and also interferes with the ergosterol synthesis and its functioning. Eventually, these processes stimulate oxidative stress, leading to inhibition of spore germination and instigating cell death [83,84,85].

### 2.6. Antibacterial Activity of Red Currant Aqueous Extracts and Rc-AgNPs

The antibacterial activity of Rc-AgNPs was assessed against human pathogenic bacteria. All the screened bacterial isolates (*Escherichia coli*, *Pseudomonas aeruginosa*, *Bacillus subtilis*, *Staphylococcus aureus*) displayed significant clear zones of inhibition (ZIs), as depicted in Figure 10A–D. *Pseudomonas aeruginosa* displayed a maximum ZI of 22 ± 0.75 mm, followed by *Staphylococcus aureus* (20 ± 0.50 mm), *Bacillus subtilis* (15 ± 1 mmm), and *Escherichia coli* (18 ± 1.25 mm) (Figure 11). The bacterial test isolates treated with Rc extract also showed antibacterial activity, but ZIs were much smaller in diameter than those exhibited by Rc-AgNP, indicating that Rc-AgNPs were more potent in inhibiting the bacterial pathogens than was the extract. However, none of the bacterial isolates screened showed inhibition by AgNO_3_.

Similarly, a previous report has shown that AgNPs synthesised from black currant pomace were highly effective in inhibiting the growth of *Escherichia coli* [28,61]. Gold nanoparticles synthesised from black currant juice were tested against *Escherichia coli*, *Pseudomonas aeruginosa*, and *Staphylococcus aureus*. All the bacteria tested were inhibited by MIC values ranging from 13 to 26 µg mL^−^^1^ [79]. Juices derived from red currant, raspberry, cranberry, and black currant showed significant inhibition zones against all the oral pathogenic bacteria screened, which included *Streptococcus gordonii*, *Streptococcus mutans*, *Streptococcus sobrinus*, *Aggregatibacter actinomycetemcomitans*, *Actinomyces naeslundii*, *Enterococcus faecalis*, *Porphyromonas gingivalis*, and *Fusobacterium nucleatum*. Among all the juices, black currant juice caused the largest zone of inhibition, which was larger than the positive control, chlorohexidine (0.2%) [86]. Seven species of Ribes (leaves and branches) were extracted with water and different organic solvents and screened against Gram-negative and positive strains (*Klebsiella pneumoniae*, *Escherichia coli*, *Enterococcus faecalis*, and *Staphylococcus aureus*). *Staphylococcus aureus* showed lower minimum inhibitory concentration values (15 to 500 μg mL^−^^1^), followed by *Enterococcus faecalis* (62 to 500 μg mL^−^^1^). *Klebsiella pneumoniae* and *Escherichia coli*, on the other hand, did not show any inhibitory activity [80].

The Rc-AgNPs exhibited robust antibacterial activity against both Gram-positive and negative bacteria in this study. The significant growth inhibition of bacterial isolates caused by Rc-AgNPs indicated that the synthesised nanoparticles were able to penetrate the cells and cause damage to vital organelles and biomolecules, interrupting their growth. Though some of the aforementioned studies have shown poor antibacterial activity of extracts and NPs against Gram-negative bacteria; in the present study, *Pseudomonas aeruginosa* showed the highest inhibition with the largest ZI. Similar to our finding, Gram-negative bacteria were highly susceptible to AgNPs, and their inhibitory activity was attributed to their narrow cell walls, which are easier to penetrate than Gram-positive cells [87,88].

Even though the mechanisms of AgNPs against bacteria have been discussed rather extensively, the precise mode of action still needs more clarity [84]. The most acceptable views regarding the mechanism of AgNPs against microorganisms are either attributed to the silver ions or the NPs themselves [89]. Silver ions have special affinities towards certain groups, such as the sulphhydryl found in the enzymes and proteins of the cell membrane. The binding of silver ions causes protein deactivation, augmenting the cell permeability [90], resulting in an unrestricted rapid influx of silver ions, leading to disruption of the cell envelope [91,92]. Moreover, the entry of silver ions into bacterial cells damages the DNA structure, inhibits ATP generation, deactivates the respiratory enzymes, and generates ROS. Several significant cell processes such as protein synthesis, replication, and cell reproduction are disrupted, which leads to cell lysis [93,94]. The antibacterial mechanism of NPs is very similar to the antifungal mechanism explained above. Precisely, the NPs bind to the cell wall, alter its permeability, and infiltrate inside the cell. This damages the cell membrane and leads to the leakage of cellular contents, which ultimately leads to cell death [95,96].

The potent antifungal and antibacterial activity exhibited by Rc extracts in this study could be due to the numerous chemical substances present in red currants. Previous reports have identified phenolic compounds such as phenolic acids, phenolic polymers (tannins) and flavonoids (myricetin, quercetin), anthocyanin, and ascorbic acid as the main active secondary metabolites in different Ribes species, including red currants [97,98,99]. All the aforementioned secondary metabolites have been associated with antimicrobial properties exhibited by red currants, black currants, and other Ribes species [81,100,101,102]. The red currant NPs (Rc-AgNPs), however, showed stronger antimicrobial activity than the aqueous extracts of red currants. Similar findings were demonstrated in earlier studies [26,103]. During nanoparticle synthesis, the secondary metabolites present in plant extracts assist in the reduction of silver ions to NPs, accompanied by capping, which renders them more effective, biologically active, and also biocompatible [104,105,106].

Hence, based on the findings of the present study, the potent antifungal and antibacterial activity exhibited by Rc-AgNPs could be attributed to the small-sized spherical AgNPs and the varied bioactive compounds witnessed in the FTIR studies. Secondary metabolites found in red currant extract, such as phenols, tannins, terpenoids, polyphenols, and flavonoids, assist in capping, reduction, providing stability to NPs, and preventing agglomeration [73,74,75]. Furthermore, red currants contain phenols, tannins, terpenoids, polyphenols, and flavonoids (quercetin, myricetin, chlorogenic acid, ferulic acid), all of which possess antimicrobial properties [100,107].

## 3. Materials and Methods

### 3.1. Plant Material and Chemicals

Red currants (berries) were purchased from a supermarket in Riyadh, Saudi Arabia. Culture media (Mueller–Hinton agar, potato dextrose agar) and silver nitrate were purchased from Sigma Aldrich-Merck KGaA, Darmstadt, Germany. All the chemicals were of analytical grade, with 99% purity.

### 3.2. Microorganisms

Several phytopathogenic fungi and some bacteria were selected for the antifungal and antibacterial assays. The fungi chosen for the study were *Alternaria alternata*, *Botrytis cinerea*, *Colletotrichum musae*, *Fusarium oxysporum*, *Pestalotiopsis mangiferae*, and *Trichoderma harzianum*. Bacterial isolates chosen were *Bacillus subtilis*, *Escherichia coli*, *Staphylococcus aureus*, and *Pseudomonas aeruginosa*. All the bacterial isolates used in this study were obtained from the King Khalid University Hospital, Riyadh, Saudi Arabia, while the plant pathogenic fungi were procured from the College of Food and Agricultural Sciences, Department of Plant Protection, King Saud University, Riyadh, Saudi Arabia.

### 3.3. Preparation of Red Currant (Rc) Extract

The red currants (berries) were washed thoroughly. The aqueous extract was prepared by adding 10 g of fresh pulverised berries to 100 mL of distilled water [32]. The berries were crushed in a mortar and pestle. This mixture was heated (60 °C) for 20 min, and after 24 h of incubation at 25 °C, the mixture was filtered (Whatman’s filter paper, No. 1). The filtrate was subjected to centrifugation at 5000 rpm for 10 min. After centrifugation, the clear supernatant was used for the experimental studies and for the synthesis of AgNPs.

### 3.4. Synthesis of Silver Nanoparticles Using Aqueous Extract of Rc

The synthesis of AgNPs was carried out by following the method of Kumar et al. [51]. A silver nitrate solution (AgNO_3_) of 1 mM was prepared by adding silver nitrate (powder) to a fixed volume of distilled water. Precisely, the reaction mixture was prepared by adding aqueous Rc extract (5 mL) to silver nitrate solution (45 mL) in a glass container. This reaction mixture was kept in direct sunlight and shaken at regular intervals. The colour change in the mixture from its original light pink to brown was closely monitored. The time taken for the change in colour was noted, as it marked the end of the reaction with the formation of red currant silver nanoparticles (Rc-AgNPs).

### 3.5. Characterisation of Rc-AgNPs

#### 3.5.1. UV–Vis Spectroscopic Analysis of Rc-AgNPs

The synthesis of Rc-AgNPs was indicated by colour change. After the colour stabilised, the Rc-AgNPs solution was analysed by a UV–Vis spectrophotometer (Shimadzu, Japan-model No-1800, Kyoto, Japan).

#### 3.5.2. Fourier Transform Infrared Spectroscopy

The Rc-AgNPs were screened with a Fourier transform infrared spectroscope (FTIR) to inspect the functional groups of the bioactive compounds present in the red currant extract and the Rc-AgNPs. The spectrum with several bands was obtained using a KBr pellet in the scan range of 400–4000 cm^−^^1^ (FTIR-Thermo Scientific, Waltham, MS, USA, Model-Nicolet-6700).

#### 3.5.3. Energy-Dispersive X-ray Analysis with Field-Emission Scanning Electron Microscopy

The elemental analysis of Rc-AgNPs was carried out with FE-SEM (field-emission scanning electron microscopy—model JSM-7610F-Japan) coupled with an EDX detector. An energy-dispersive X-ray (EDX) spectrum was obtained (30 kV). The spectrum depicts the presence of silver and various other elements.

#### 3.5.4. The Dynamic Light Scattering Analysis

The size distribution of Rc-AgNPs was determined with a dynamic light scattering analyser (DLS). The obtained spectrum provides the hydrodynamic size, distribution, and polydiversity index (PDI). The spectrum was captured on a DLS analyser model: Nano Series-Zeta Sizer-ZEN-3600, Malvern, UK.

#### 3.5.5. Transmission Electron Microscopy

The transmission electron microscope (TEM) allows us to precisely determine the morphology (size, shape) of the nanoparticles. The microphotographs of Rc-AgNPs were captured on the latest model of TEM-JEOL JEM-Plus-1400, Tokyo, Japan.

The preparation of all the samples for the above-mentioned analysis was performed by following and abiding by the manufacturer’s instructions.

### 3.6. Antifungal Activity

The antifungal activity of the Rc-AgNP and Rc extract was assessed against a panel of phytopathogenic fungi by a previously described method [32]. To obtain pure and fresh fungal cultures, test fungi were subcultured on potato dextrose agar (PDA) and used for antifungal studies after 7 days of growth. A disc (6 mm) was removed and cut off with a sterile borer from the subcultured fungal plates of different test fungi and used for the assay. Fresh PDA was prepared and amended either with 500 mL of the aqueous Rc-extract, synthesised Rc-AgNPs, or with AgNO_3_ separately. The amended media was carefully and aseptically poured into sterile PDA plates and allowed to cool at room temperature (25 °C). Once the amended media in the plates solidified, the 6 mm disc of test fungi was placed upside down in the centre of each plate and kept at 28 °C for 9 days. The diameter of the mycelial growth was measured and the percentage inhibition was calculated [32]. Positive control was the mixture of the fungicide carbendazim (0.2%). The antifungal assay with each fungal isolate was carried out in a similar manner, and each experiment was run in triplicate.

### 3.7. Antibacterial Activity

The effect of aqueous extract of red currant, Rc-AgNPs, and AgNO_3_ solution on the growth of the bacterial test isolates was evaluated by the well-diffusion method [32]. Bacterial cultures of all the test isolates were grown overnight (24 h) in nutrient broth. A working suspension of bacterial test isolates was prepared by adjusting the concentration to ∼106 CFU per mL (colony-forming unit), which corresponded to 0.5 Mac Farland and was used for the assay. Nutrient agar (NA) plates were smeared evenly with 100 µL of bacterial suspension and allowed to solidify. After solidification, wells were punched (4 mm) in the agar in each plate, and each well was filled with either of the test solutions as mentioned, i.e., Rc extract, Rc-AgNPs, or AgNO_3_ (solution). The plates were then incubated for 24 h at 37 °C. After the incubation period, the clear zone around each well was measured and the measurements recorded. The clear zones are the zones of inhibition (ZIs) with no bacterial growth. Each bacterial test isolate was screened in a similar manner with Rc extract, Rc-AgNPs, AgNO_3_, and with the antibiotic disc (30 µg) (positive control). The experiment was repeated three times with each bacterial isolate.

### 3.8. Statistical Analysis

All of the values and data displayed in figures in the present study were run in triplicates (±SD). Analysis of variance (ANOVA) and Tukey’s HSD tests were implemented for significant differences (*p* ≤ 0.05). The statistical tests were run on GraphPad prism version 8.4.3.686. and XLSTAT (software version-2020).

## 4. Conclusions

Red currants are an excellent source of antioxidants (phenolic and flavonoids) and are rich in a wide variety of health-benefiting minerals, vitamins, and organic acids. The biosynthesised nanoparticles from red currants exhibited potent antibacterial and antifungal activity. Based on the findings of this study, the Rc-AgNPs could be employed in the formulation of effective antifungal and antibacterial drugs to combat the increasingly resistant strains. However, future research to understand their mechanism and identify the compounds in the extracts will help widen their utilisation in various biomedical applications in pharmaceuticals, agroindustry, and medical therapeutics.

## Figures and Tables

**Figure 1 molecules-27-02186-f001:**
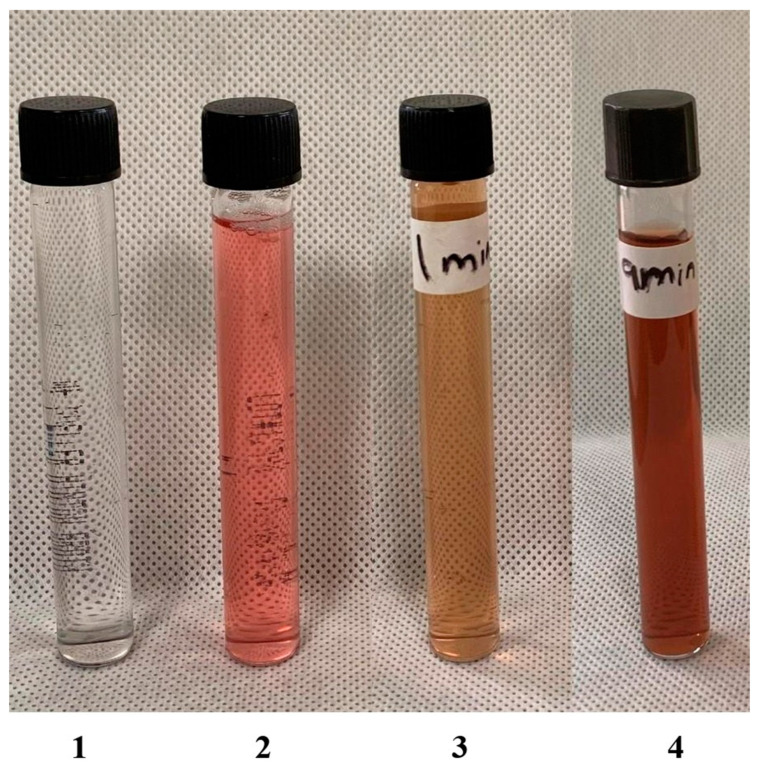
The formation of AgNPs using aqueous extracts of red currants (berries): **1**—silver nitrate solution (AgNO_3_); **2**—red currant aqueous extract; **3**—reaction mixture after 1 min of exposure to sunlight; **4**—formation red currant nanoparticles (Rc-AgNPs) after 9 min.

**Figure 2 molecules-27-02186-f002:**
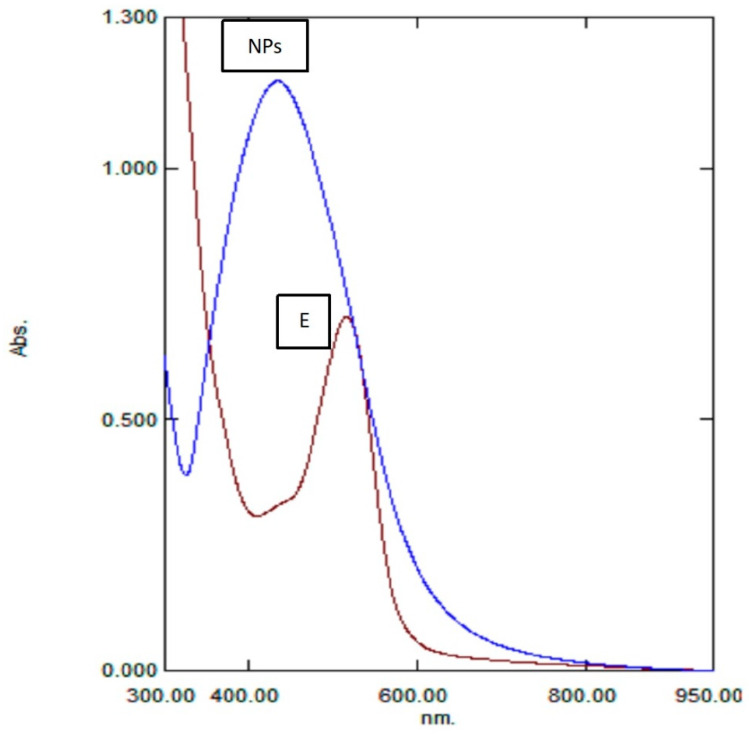
UV–Vis spectrum showing the LSPR peak of synthesised Rc-AgNPs at 435 nm. E-red currant aqueous extract, NPs-Rc-AgNPs.

**Figure 3 molecules-27-02186-f003:**
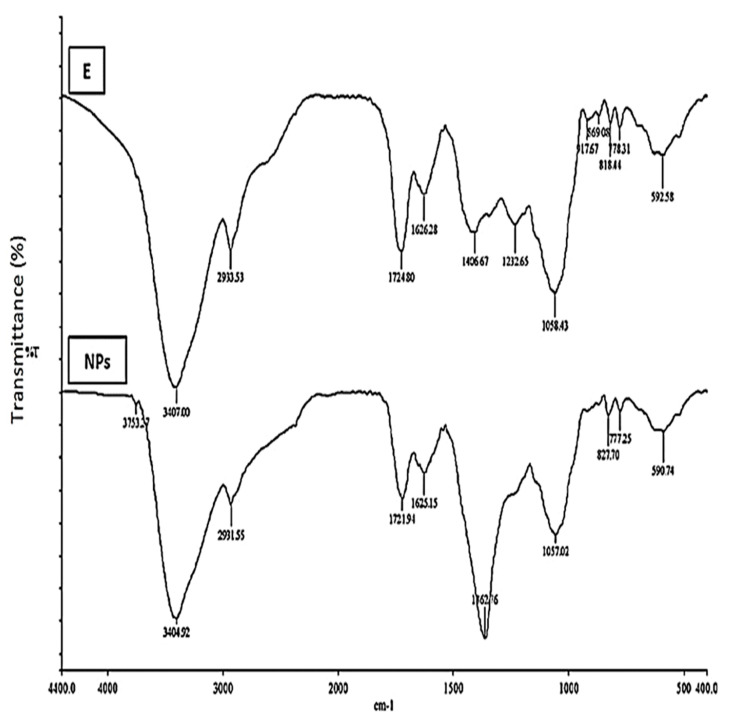
The Fourier transform infrared (FTIR) spectrum of red currant AgNPs (NPs) and red currant aqueous extract (E). Several peaks denoting the functional groups of bioactive compounds were obtained in a range of 500–4000 cm^−1^ on a spectrometer (Nicolet).

**Figure 4 molecules-27-02186-f004:**
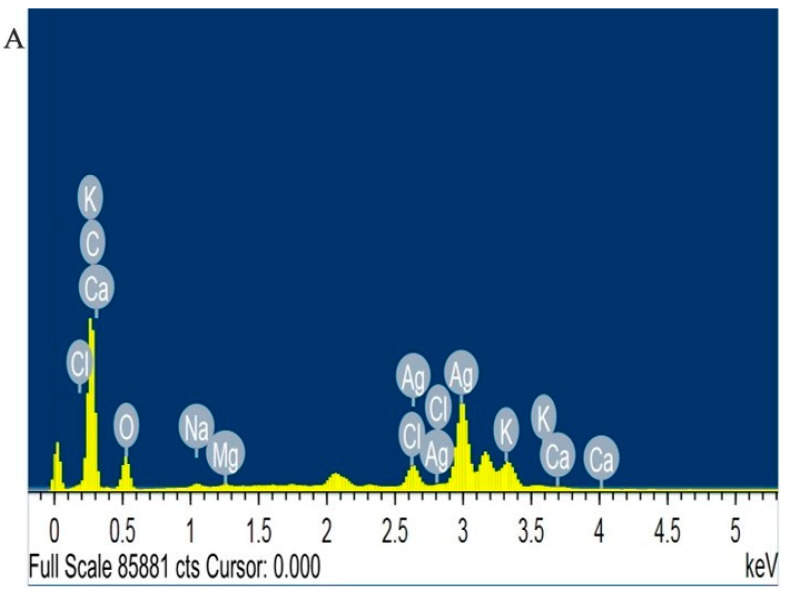
Energy-dispersive X-ray (EDX) spectrum of red currant AgNPs: (**A**) peak related to silver at 3 KeV; (**B**) elemental composition of the red currant AgNPs.

**Figure 5 molecules-27-02186-f005:**
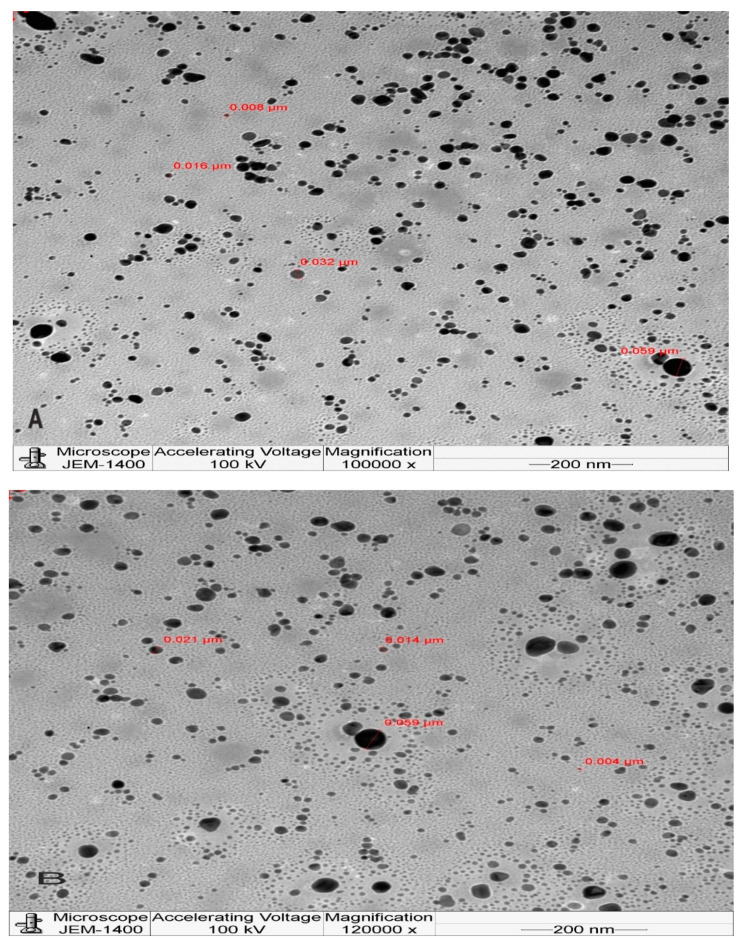
Transmission electron microphotographs of red currant AgNPs: (**A**,**B**) depicts the small size (4 to 59 nm) and spherical of AgNPs synthesised from red currant aqueous extracts.

**Figure 6 molecules-27-02186-f006:**
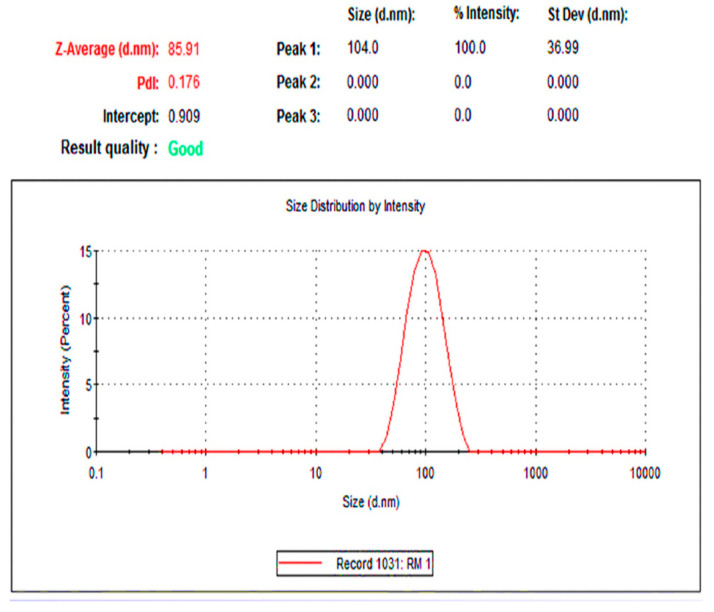
The spectrum obtained by the dynamic light scattering analysis of red currant-AgNPs. The spectrum shows the hydrodynamic diameter (Z-average-d.nm); PDI, polydispersity index.

**Figure 7 molecules-27-02186-f007:**
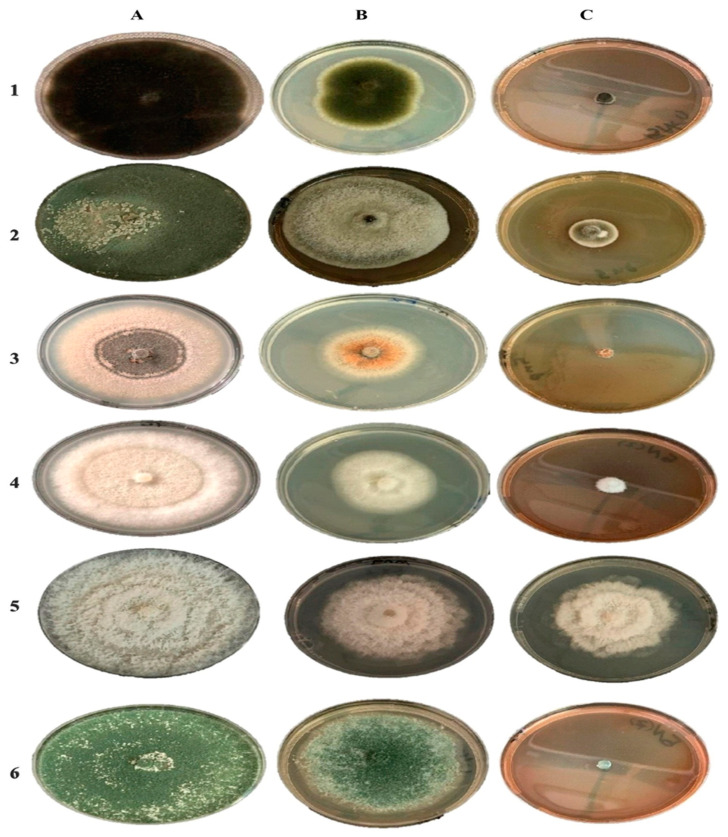
Antifungal activity of the red currant aqueous extracts and red currant-AgNPs against a panel of plant pathogenic fungi: (**A**) control, not treated; (**B**) treated with aqueous extracts of red currants; (**C**) treated with red currant-AgNPs. **1**—*Alternaria alternata*; **2**—*Botrytis cinerea*; **3**—*Colletotrichum musae*; **4**—*Fusarium oxysporum*; **5**—*Pestalotiopsis mangiferae*; **6**—*Trichoderma harzianum*.

**Figure 8 molecules-27-02186-f008:**
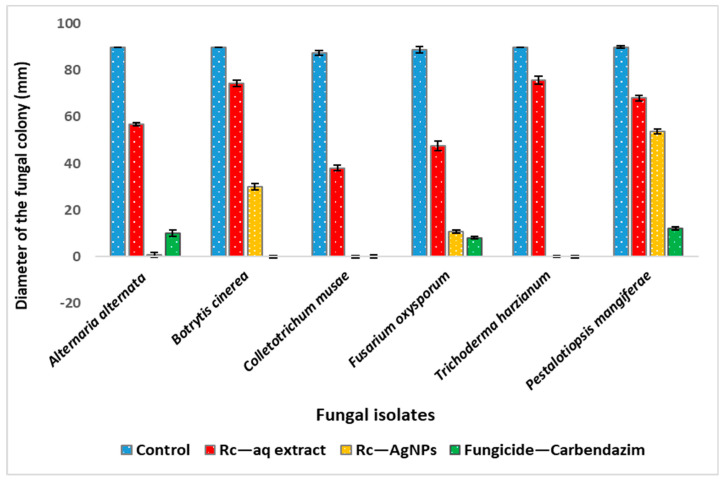
Diameter of the mycelial growth of fungal phytopathogens treated with red currant aqueous extracts, red currant-AgNPs, and the fungicide. The values depicted in the graph are an average (±SD) of three independent experimental replicates. Significant differences in means (*p* ≤ 0.05) were determined by analysis of variance ANOVA and Turkey HSD.

**Figure 9 molecules-27-02186-f009:**
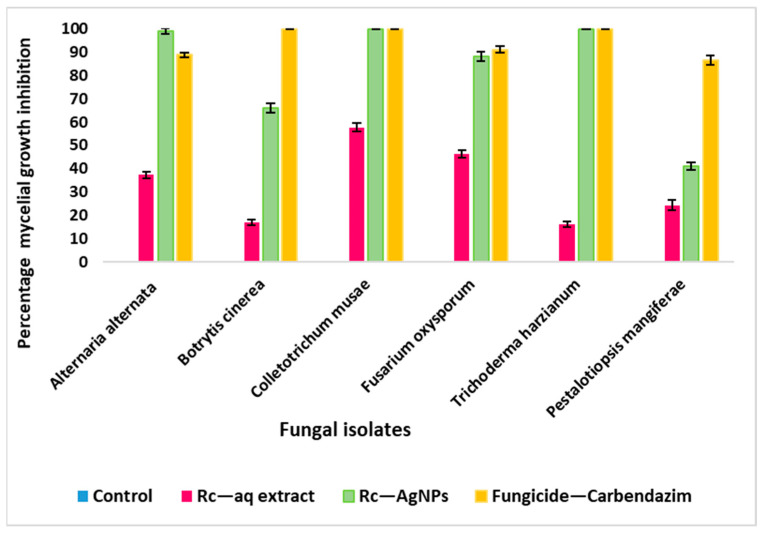
Percentage of mycelial growth inhibition of some fungal phytopathogens treated with red currant aqueous extracts, red currant-AgNPs, and the fungicide carbendazim (2%). The values displayed in the above graph are means of triplicate experimental replicates (±SD). Significant differences in means (*p* ≤ 0.05) were determined by analysis of variance ANOVA and Turkey HSD.

**Figure 10 molecules-27-02186-f010:**
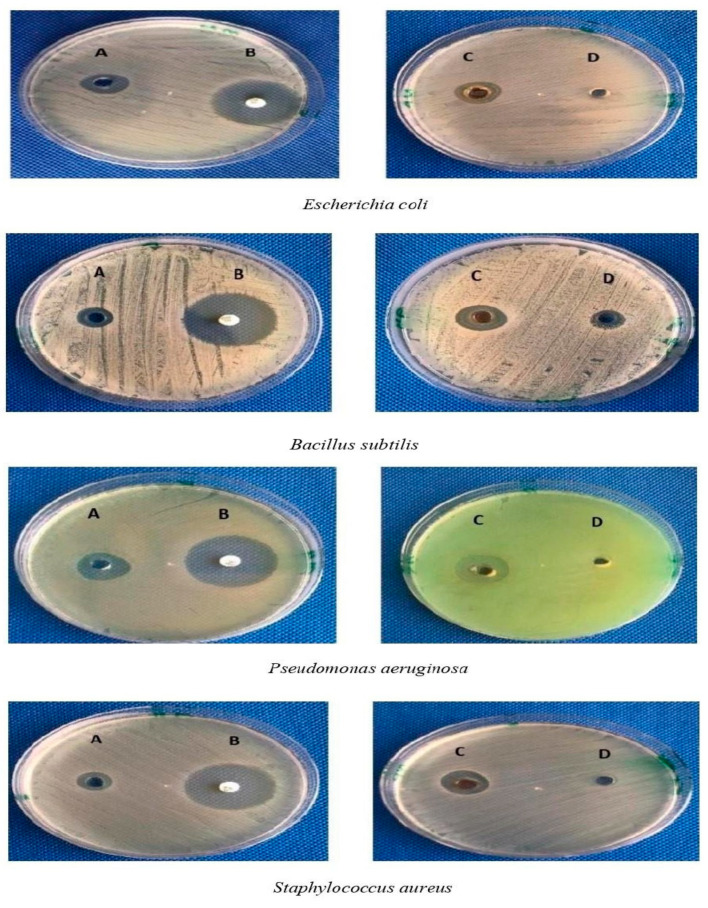
Antibacterial activity of A—red currant aqueous extract; B—antibiotic (Gentamycin-30 µg); C—red currant-AgNPs, D—AgNO_3_ against pathogenic bacterial isolates.

**Figure 11 molecules-27-02186-f011:**
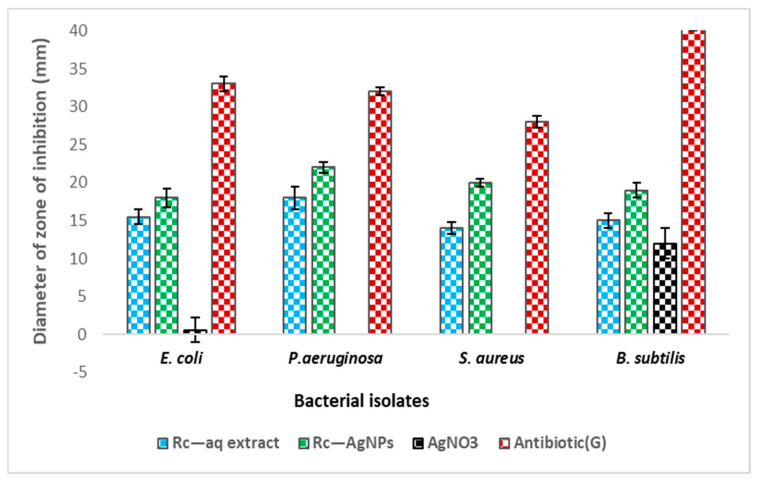
Effect of various treatments (red currant aqueous extract; red currant-AgNPs; AgNO_3_) and antibiotic (Gentamycin-30 µg) on the growth of Gram-positive and negative bacterial test isolates. Agar well-diffusion assay was conducted, and zone of inhibition (clear zones-mm) around each well was measured. The values in the experiment are means of three replicates run for each set (SD±). Significant differences in means (*p* ≤ 0.05) were determined by analysis of variance ANOVA and Turkey HSD. *E. coli—Escherichia coli; P. aeruginosa—Pseudomonas aeruginosa; S.aureus—Staphylococcus aureus; B.subtilis**—**Bacillus subtilis*.

## Data Availability

All the data of the present study are present in the manuscript.

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
