# Peer review of "Sunlight-Mediated Green Synthesis of Silver Nanoparticles Using the Berries of Ribes rubrum (Red Currants): Characterisation and Evaluation of Their Antifungal and Antibacterial Activities"

_molecules, 2022, doi:10.3390/molecules27072186_

Round 1
Reviewer 1 Report
The study of the synthesis of silver nanoparticles using the berries of Ribes rubrum (red currants) is a good contribution on green chemistry for the synthesis of Ag nanoparticles through the use of an accessible method, with a non-explored plant material. The paper contains interesting and original results that should be published, after checking up some suggestions and minor corrections.
- 3 line 135 the the
- 5 line 196 "Energy dispersive X-ray analysis coupled with SEM revealed the elemental composition" is SEM or TEM?
- 5 line 216 it is mentioned the size of the nanoparticles between 8 nm to 59 nm, but in P. 11 footnote of Fig. 5 says 4 nm to 59 nm (line 361)
- 6 line 252 defin MIC and MFC in the text
P.4 Line 151-155 in the first part of the results, it is discussed that "when nano synthesis is carried out in containers made of glass or plastic, the transmission of UV light is totally diminished as UV light cannot pass through them. Hence, blue light in the visible region of the light spectrum plays a major role in the photo reduction of silver during nano synthesis” This observation seems to contrast with the reduction of the nanoparticles reported in the present paper, where glass vials were used for making the sunlight-mediated synthesis. Please explain the relation of this statement and the glass material used in this work?
In the study of antifungal and antibacterial activity against different fungal or bacteria, always the evaluations are compared in Rc-aq-extract, I would suggest that in some part of the discussion to explain the proposal of these comparisons (Rc-AgNPs and Rc-aq-extract), since Rc-aq-extract does not contain any silver nanoparticles?
Reviewer 2 Report
Review of molecules-1644420
This manuscript describes the biosynthesis of silver nanoparticles (AgNPs) from silver nitrate with the assistance of red currant extract. It is claimed that the reaction only takes place for 9 minutes only, under sunlight. The prepared AgNPs are tested against several fungi and several bacteria to show the potential antifungal and antibacterial properties, respectively. The characterizations are complete. However, there are many items to be corrected, before it is considered for acceptance, as follows:
- Because of the potential of generating confusion, please do not abbreviate the scientific names (Latin names). Please write the complete genus.
For example:
- Abbreviated genus “P.” can be (mis)interpreted as either Pseudomonas, or Pestalotiopsis.
- Abbreviated genus “S.” can be (mis)interpreted as Staphylococcus, or Streptococcus. In the line 299, there are S. mutans, S. sobrinus, and later only six lines below, there is S. aureus.
- Abbreviated genus “E.” can be (mis)interpreted as Escherichia, can be Enterococcus. An annoying example in line 306-307: E. faecalis followed by E. coli.
Not all the readers of this journal are expert in microbiology, and this overuse of abbreviation may mislead the readers, and they may read it as the WRONG names of “Escherichia faecalis” or “Enterococcus coli”, or even there is a possibility to interpret S. aureus as “Sphingomonas aureus”, or P. aeruginosa as “Paramecium aeruginosa”, where all of them are wrong.
--> Therefore, please write ALL genus in complete way, NOT in the abbreviated style, and in ITALIC.
- In addition, not all of the scientific names are consistently written in italic in this manuscript, please correct them.
- Please check again the journal requirement about the flow of the manuscript. Does it stick to the classic IMRAD (Introduction, Methods, Results and Discussion with figures displayed one by one before a description)? Or, does it allow for the flow of Introduction, then directly jumps to Results and Discussion, get back to Methods, and finally Figures? Please recheck.
- Line 115-792 (Section 2, 3, 4, References): Please use the Justified alignment.
- There are some references to add to show about recent biosynthesis of silver nanoparticles:
- Food and Bioproducts Processing 121 (2020) 193-201 https://doi.org/10.1016/j.fbp.2020.02.008
- International Journal of Biological Macromolecules 140 (2019) 168-176 https://doi.org/10.1016/j.ijbiomac.2019.08.131
- Please be consistent in writing units in the format of X Y-1, not X/Y. The wavenumber of FTIR is commonly written as cm-1, and almost rarely written as “/cm” although it is principally correct. So, please use µg mL-1, mg mL-1.
- Please use subscripted 3 for writing chemical structure, e.g. AgNO3. Please correct it.
- Line 5: Last author’s surname must be started with uppercase A
- Line 31-32 (Keywords): …antifungal activity, antibacterial activity --> please make it as two separated keywords.
- Line 83-84: …Gram-positive and Gram-negative --> please use uppercase G letter, because it is derived from the name of Hans Christian Gram, the iconic microbiologist known for Gram staining.
- Line 88-98: Please add some references in this paragraph.
- Line 97: Scientific names must be written in italic.
- Line 121: Please use subscripted 3 for writing chemical structure, which is AgNO3 (there are two in this line).
- Line 142: Please use subscripted 3 for writing chemical structure of AgNO3.
- Line 156-176 (Section 2.2), please show the Ag peak(s).
- Line 166: Please use subscripted -1 for the unit of wavenumber, cm-1.
- Line 210-234 (section 2.4): Do not write all the lines in this paragraph in italic.
- Line 228-229: …cannot be corroborated, and the measurement by DLS reported a generally larger size than those measured by TEM.
- Line 237-245: Scientific names must be written in italic, in complete and not abbreviated genus. Write Pestalotiopsis mangiferae, Alternaria alternate, Botyris cinerea, Colletotrichum musae, Fusarium oxysporum, Colletotrichum musae, Fusarium oxysporum, Bacillus subtilis, Escherichia coli, Staphylococcus aureus,Pseudomonas aeruginosa, Klebsiella pneumoniae must be written ended with e.
- Line 247: Please use subscripted 3 for writing chemical structure of AgNO3
- Line 251: Scientific names must be written in italic, in complete and not abbreviated genus.
- Line 252: Please be consistent in writing units in the format of X Y-1, not X/Y.
- Line 254: Please be consistent in writing units in the format of X Y-1, not X/Y.
- Line 260: Please be consistent in writing units in the format of X Y-1, not X/Y.
- Line 261: Please be consistent in writing units in the format of X Y-1, not X/Y.
- Line 258: Scientific names must be written in italic, in complete and not abbreviated genus.
- Line 259: Scientific names must be written in italic, in complete and not abbreviated genus.
- Line 260: Scientific names must be written in italic, in complete and not abbreviated genus.
- Line 261: Scientific names must be written in italic, in complete and not abbreviated genus.
- Line 285, 286, 287, 288: Scientific names must be written in italic, in complete and not abbreviated genus.
- Line 289: Please use subscripted 3 for writing chemical structure of AgNO3
- Line 294 vs 295: First, there is the abbreviated E. coli, and then followed by the unabbreviated Escherichia coli. This is not consistent. Therefore, to erase doubt, write all scientific names in italic, in complete and not abbreviated genus.
- Line 297: Please be consistent in writing units in the format of X Y-1, not X/Y.
- Line 299: Scientific names must be written in italic, in complete and not abbreviated genus.
- Line 305: Klebsiella pneumoniae --> ended with e.
- Line 306: Scientific names must be written in italic, in complete and not abbreviated genus.
- Line 306: Please be consistent in writing units in the format of X Y-1, not X/Y.
- Line 313: Scientific names must be written in italic, in complete and not abbreviated genus.
- Line 314: …finding Gram-negative.. --> Gram, with uppercase G
- Line 342: Check if section 2.7 Figures is allowed by the style of this journal.
- Figure 1: Please rearrange the figures to be 1-2-3-4 from left to right, for the transparent, to the pink, to the brownish orange, and finally to the dark red solution.
- Caption of Figure 3: 4000 cm-1 --> delete “/”, because the unit is NOT “/cm-1”.
- Figure 4a: Explain what is the sharp tall peak located at 1.8 keV.
- Figure 5a and 5b: Please rearrange the two figures to be in the same alignment.
- Figure 7: Nice and detailed figures, but please resize those 18 figures to fit one page (and the figure caption must also placed in the same page as the 18 figures.
- Caption of Figure 8: Significant difference is p <0.05, not p >0.05.
- Caption of Figure 9: Significant difference is p <0.05, not p >0.05.
- Caption of Figure 10: Please use subscripted 3 for writing chemical structure of AgNO3
- Caption of Figure 11: Significant difference is p <0.05, not p >0.05.
- Line 421: Please use subscripted 3 for writing chemical structure of AgNO3
- Line 462: Please use subscripted 3 for writing chemical structure of AgNO3
- Line 464-465: Please write the correct degree sign for the unit ° Do not use superscripted zero, lowercase o, or uppercase O.
- Line 475: …106 CFU per mL …--> not “per mL CFU).
- Line 479: Please use subscripted 3 for writing chemical structure of AgNO3
- Line 480: Please write the correct degree sign for the unit ° Do not use superscripted zero, lowercase o, or uppercase O.
- Line 483: Please use subscripted 3 for writing chemical structure of AgNO3
- Line 488: GraphPad Prism --> this is a brand, so we must follow the style of writing of the brand, which is with uppercase P letters, and with “Graph” and “Pad” combined together as “GraphPad”.
- References: Please follow the font type, the font size of the style of this journal. Write all scientific names in the title of the cited references in italic!
- Line 532, 535, 538, 539, 547, 557, 560, 563, 565, 574, 586, 594, 607, 610, 611, 613, 627, 630, 643, 645, 648, 652, 656, 659, 663, 664, 671, 680, 690, 694, 697, 698, 713, 714, 735, 741, 744, 747, 753, 771, 774, 775, 781,
- References: There are journal names that written in italic, there are that not written in italic. Please be consistent!
- Line 556: Nanotechnology, not abbreviated, not separated.
- Reference 35:??? Very confusing way to write a reference. Please revise.
- Line 633: Acta Bibl.
- Line 684: Please use subscripted 2 and 2 for writing chemical structure of H2O2.
- Line 694: ciniformis with lowercase c.
- Line 706: J. Food Drug Anal. --> uppercase F, D, A
- Line 709: J Phys… --> with s
- Line 724: J. Coll. Interface Sci --> uppercase C, I, S.
- Line 735: Arab J. Chem. --> separate the word “ArabJ” with a space.
- Line 762: Biochim. Biophys. Acta --> uppercase B, B, A.
- Line 766: Int. J. Nanomed. --> uppercase N.
- Line 773: Please use superscripted + when writing the Ag+
- Line 776: J Phys….--> with s
Author Response
Please see the attachment."

Round 2
Reviewer 2 Report
Review of molecules-1644420-v2
The authors have addressed all issues very well. The manuscript can be accepted for publication.
Note: There are some items that must be amended during the proofreading stage, as follows:
- Please use justified style for all paragraph in this manuscript.
- Line 159: Please write Ag+ with superscripted +
- Line 161: Please write Ag+ with superscripted +
- Section 2.2: Sorry for unclear suggestion. I asked to display Ag-related functional groups, because the displayed results are of common functional groups such as C=O, CH, OH, etc. Maybe it can be displayed via XPS (X-ray photoelectron spectroscopy). But in my opinion, the EDX result has represented quite well about the chemical characteristics of the materials in this submission. Therefore, there is no need to revise section 2.2 (except for using the justified style to tidy up the paragraphs).